*Report*

# Inflating bacterial cells by increased protein synthesis

Markus Basan[1,*,†], Manlu Zhu[2,3,†], Xiongfeng Dai[2,3], Mya Warren[2], Daniel Sévin[1], Yi-Ping Wang[3] & Terence Hwa[2,4,**]

## Abstract

Understanding how the homeostasis of cellular size and composition is accomplished by different organisms is an outstanding challenge in biology. For exponentially growing *Escherichia coli* cells, it is long known that the size of cells exhibits a strong positive relation with their growth rates in different nutrient conditions. Here, we characterized cell sizes in a set of orthogonal growth limitations. We report that cell size and mass exhibit positive or negative dependences with growth rate depending on the growth limitation applied. In particular, synthesizing large amounts of "useless" proteins led to an inversion of the canonical, positive relation, with slow growing cells enlarged 7- to 8-fold compared to cells growing at similar rates under nutrient limitation. Strikingly, this increase in cell size was accompanied by a 3- to 4-fold increase in cellular DNA content at slow growth, reaching up to an amount equivalent to ~8 chromosomes per cell. Despite drastic changes in cell mass and macromolecular composition, cellular dry mass density remained constant. Our findings reveal an important role of protein synthesis in cell division control.

**Keywords** cell size; cell division; cellular DNA; cell volume; growth rate

**Subject Categories** Metabolism; Protein Biosynthesis & Quality Control; Quantitative Biology & Dynamical Systems

**Mol Syst Biol. (2015) 11: 836**

## Introduction

Throughout biology populations of growing cells are able to achieve robust coordination of biomass production with cell volume expansion and cell division, often resulting in tight control of cell size and cellular composition. The growth rate dependence of cell size has long been known under different nutrient conditions in the model organism *Escherichia coli* (Schaechter *et al*, 1958; Volkmer & Heinemann, 2011; Hill *et al*, 2012) and other microbes (Di Talia *et al*, 2009; Turner *et al*, 2012; Soifer & Barkai, 2014), but the origin underlying this relations remains unknown. Recently, the addition of an approximately constant cellular mass per cell division, as a heuristic mechanism for stable cell size regulation, has been supported with substantial experimental evidence in different microorganisms (Amir, 2014; Campos *et al*, 2014; Taheri-Araghi *et al*, 2015). Closely related to the question of cell size regulation is the coordination of cellular composition with growth. For *E. coli* grown in different nutrient conditions, cellular DNA content exhibits a similar growth rate dependence as cell size (Helmstetter & Cooper, 1968; Hill *et al*, 2012). But because of the tight correlation between growth rate, cell size, and DNA content, observed under this standard growth limitation, the underlying causal interrelations remain unclear.

In the present study, we describe a set of surprising findings obtained from orthogonal modes of growth limitation. These results challenge several commonly held notions about the coordination of cell size and cellular composition and highlight the role of protein synthesis in mediating cell size control.

## Results

We characterized the dependence of cell size on growth rate for three distinct modes of growth limitations (Appendix Table S1) of *E. coli* K-12 cells: limitation in nutrient uptake by different growth media, limitation in protein synthesis by antibiotics, and limitation in proteome allocation by expression of useless proteins (LacZ), following recent quantitative studies of bacterial physiology (Scott *et al*, 2010; You *et al*, 2013; Hui *et al*, 2015). All samples were taken from exponentially growing cultures (Appendix Fig S1). In each case, the size of cells was determined via microscopy and automated image analysis (see Materials and Methods). Remarkably, cell sizes obtained for these three distinct limitations strongly diverged at comparable growth rates, as illustrated by snapshots of cells collected from cultures at similar $OD_{600}$ (Fig 1A), with the size distributions shown in Fig 1B and Appendix Fig S2, with the means and variances of all conditions reported in Appendix Table S2. The

1  Institute of Molecular Systems Biology, ETH Zürich, Zürich, Switzerland
2  Department of Physics, University of California at San Diego, La Jolla, CA, USA
3  State Key Laboratory of Protein and Plant Gene Research, School of Life Sciences, Peking University, Beijing, China
4  Institute for Theoretical Studies, ETH Zürich, Zürich, Switzerland
   *Corresponding author. Tel: +41 44 633 40 52; E-mail: basan@imsb.biol.ethz.ch
   **Corresponding author. Tel: +1 858 534 7263; E-mail: hwa@ucsd.edu
   †These authors contributed equally to this work

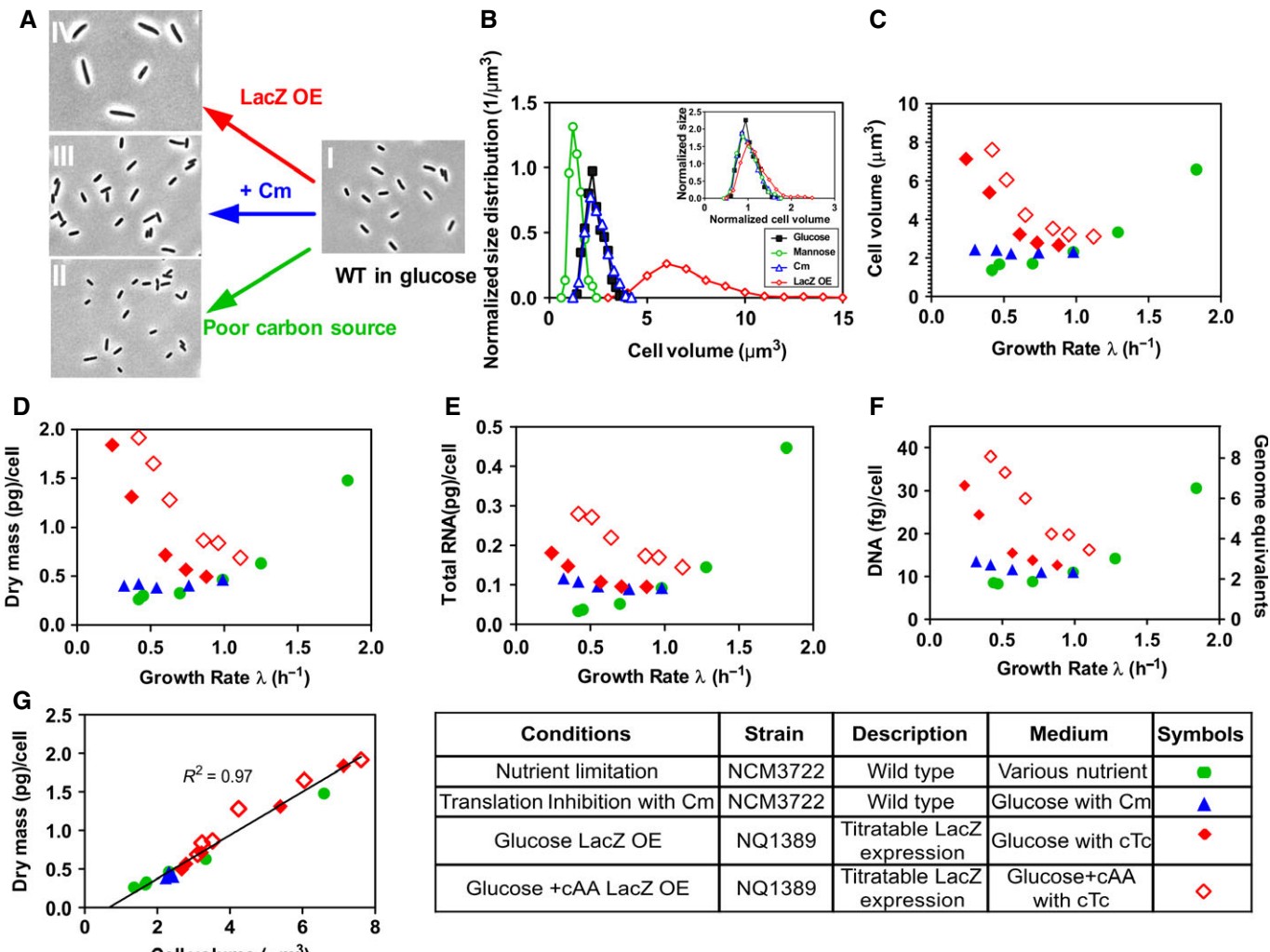

**Figure 1.  Cell size and content under different growth limitations.**

A   Snapshots of bacteria from different culture conditions at similar $OD_{600}$ (~0.4) and the same magnification: I. glucose ($\lambda \approx 0.98$/h); II. mannose ($\lambda \approx 0.41$/h); III. glucose + 8 µM Cm ($\lambda \approx 0.32$/h); IV. LacZ OE, glucose + 15 ng/ml cTc ($\lambda \approx 0.25$/h). Cultures under different growth limitations (II–IV) exhibit large differences in cell size at comparable growth rates.

B   Normalized cell size distributions, as quantified by automated image analysis, for cells taken from the conditions described in panel (A). Distributions for cells grown in mannose, Cm, and LacZ OE were taken at comparable growth rates. Inset, density distributions for cell volume normalized by average cell size. When normalized by mean cell size, the different distributions appear very similar.

C   Mean cell volume obtained under the different growth limitations plotted against the corresponding growth rate of the culture (see Appendix Table S2 for standard deviations and Appendix Table S3 for the variation between repeats and different $OD_{600}$).

D   Cellular dry mass plotted against the corresponding growth rate of the culture, for each growth perturbation. The trends of cellular dry mass closely resemble the trends exhibited by cell volume (panel C).

E   Cellular RNA plotted against the corresponding growth rate of the culture, for each growth perturbation.

F   DNA content per cell. The trends in DNA content, as confirmed by DAPI staining (Fig EV3), also closely follow the change in cell size shown in panel (C) (see Fig EV1C for the correlation plot).

G   Cellular dry mass plotted against cell volume. A tight correlation exists between these quantities under all growth limitations.

observed size distributions were highly reproducible and independent of culture density (see Appendix Table S3, Appendix Fig S2E and F). The width of the size distributions largely results from differences in mean cell size as reported in recent single-cell studies (Taheri-Araghi *et al*, 2015), with the different distribution functions collapsing when normalized by the mean cell size (see inset Fig 2B). Mean cell size (volume), plotted against the growth rate (GR) of the exponentially growing culture, showed

distinct trends for the three limitations (Fig 1C, with cell length and width presented in Appendix Fig S3, and their values listed in Appendix Table S2). While cell size decreased with nutrient limitation (green circles) in accordance with previous studies (Schaechter *et al*, 1958; Volkmer & Heinemann, 2011; Hill *et al*, 2012; Chien *et al*, 2012), it remained constant under sub-lethal doses of the translational inhibitor chloramphenicol (Cm, blue triangles) and increased strongly when growth was limited by the overexpression

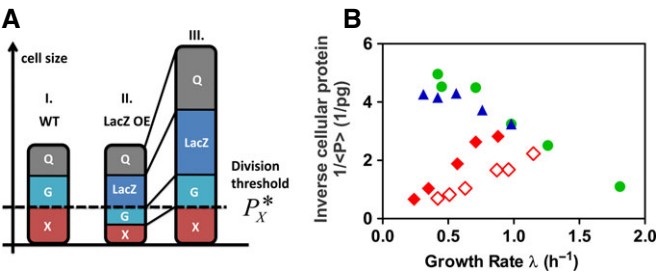

**Figure 2. The threshold initiation model of cell size control.**

A   Schematic of the initiation model. This model assumes that a threshold amount of the cell division protein X per cell, $P_X^*$, is required to trigger cell division: When the abundance of the protein X reaches a threshold level (represented by the dashed line), the cell divides at this size (left, cell I). In the LacZ OE strain, LacZ "compresses" the proteome fractions of X and other GR-dependent proteins G, while a certain fraction of the proteome Q remains constant and cannot be reduced (Hui *et al*, 2015). Hence, with LacZ OE, a cell at the size of cell I would contain a smaller amount of protein X, reducing it below the threshold level $P_X^*$, needed for cell division (middle, cell II). The cell would continue to grow and eventually divide when the cellular abundance of protein X reaches this threshold level (right, cell III). Due to the smaller proteome fraction of X under LacZ OE, a much larger cell is produced (see Box 1 for a quantitative analysis).

B   Inverse of the average cellular protein content, $1/\langle P\rangle$, versus the growth rate for the different growth limitations; same symbols as in Fig 1. According to the threshold initiation model, the plotted quantity reflects the growth rate dependence of the abundance of the cell division protein X under each mode of growth limitation (see Box 1).

(OE) of a useless protein, LacZ (red diamonds), via a linearly inducible genetic construct (see Appendix Fig S4). Indeed, Fig 1C shows that slow growing cells due to LacZ OE exhibited sizes larger than even the largest cells observed for the fastest growing wild-type cells cultured in rich media.

We also characterized the macromolecular content of the culture, namely protein, RNA, and DNA, as well as the total dry mass and cell count (Appendix Table S4). The sum of protein, RNA, and DNA was found to account for ~90% of dry mass (Appendix Fig S5), over two-third of which is protein for each of the growth limitations (Appendix Fig S6A–D). Cellular dry mass and cellular protein content, shown in Fig 1D and Appendix Fig S7, respectively, displayed quantitatively similar trends as those exhibited by the physical cell size (Fig 1C) for each growth perturbation, with a 7- to

---

**Box 1: Threshold initiator model**

The observed growth rate dependences of cell mass under the different support a simple model of cell size control. In this model, the abundance of specific cell division proteins (collectively referred to as X) in an individual cell needs to reach a threshold level in order to initiate cell division (Fantes *et al*, 1975; Wold *et al*, 1994; Boye & Nordström, 2003; Donachie & Blakely, 2003). This threshold level is defined to be constant for all growth conditions.

We denote the abundance of the division proteins X in a cell as $P_X$ and the threshold abundance as $P_X^*$. As our study is concerned with the *average* properties of the culture, we adopt a mean-field version of the above model, in which cell division takes place when the average abundance of X per cell, denoted as $\langle P_X\rangle$, reaches the threshold $P_X^*$. The quantity $\langle P_X\rangle$ is simply given by the fractional abundance of proteins X, $\phi_X$ (as a fraction of the total proteome), as $\phi_X = \langle P_X\rangle/\langle P\rangle$, where $\langle P\rangle$ is the total cellular protein $P$ averaged over the population. Note that $\phi_X$ is accessible by proteomic mass spectroscopy (Hui *et al*, 2015) if the identity of X is known.

In this mean-field model, the "size" of cells is given by the average abundance of total proteins per cell at division, denoted as $\langle P^*\rangle$. Since the cellular abundance of proteins X at division is $P_X^*$ by the definition of the model, then it follows that

$$\langle P^*\rangle = P_X^*/\phi_X. \tag{1}$$

Next, we note that the total cellular protein abundance averaged over the population of cells, and $\langle P\rangle$ is proportional to $\langle P^*\rangle$ (e.g., given by 3/4 $\langle P^*\rangle$ if cells were uniformly distributed throughout the cell cycle). Indeed, the average protein abundance per cell is seen to correlate well with the cell size (Fig EV1A). Thus, the model predicts $\langle P\rangle \propto P_X^*/\phi_X$ or

$$\phi_X \propto 1/\langle P\rangle. \tag{2}$$

Since the total cellular protein abundance is known (Appendix Fig S4 and Fig EV4), inverse of this quantity (plotted in Fig 2B) gives the GR dependence of the proteome fraction of X under the three modes of growth limitations studied.

As established in Hui *et al* (2015) and summarized in Appendix Fig S4, under LacZ OE the proteome fraction of most proteins exhibit direct proportionality to the GR ($\lambda$). Assuming that the cell division protein X follows the same trend, that is, $\phi_X = \langle P_X\rangle/\langle P\rangle \propto \lambda$, then equation (2) predicts that $1/\langle P\rangle \propto \lambda$, as verified for both LacZ OE series (filled and open red diamonds) in Fig 2B. Figure 2A gives a schematic illustration of how this mechanism would lead to the inflated cells under LacZ OE.

For nutrient-limited growth, a negative linear GR dependence is seen, that is, $1/\langle P\rangle \propto 1 - \lambda/\lambda_0$. With $\lambda_0 \approx 2.2$/h, this GR dependence corresponds to the proteome fraction reported for constitutively expressed proteins in nutrient-limited growth (Scott *et al*, 2010). While the proteome fraction of most proteins would decrease with decreasing GR under Cm inhibition as in the case of LacZ OE, since Cm inhibition results in an increased expression of ribosomal proteins with reduction in most other proteins (Hui *et al*, 2015), the data for $1/\langle P\rangle$ in Fig 2B indicate the opposite trend. The Cm inhibition data could therefore be very informative regarding the identities of cell division proteins X. Of the ~1,000 proteins quantified by quantitative mass spectroscopy analysis (Hui *et al*, 2015), the relative abundance of only a few proteins matched the profile shown in Fig 2B (blue triangles), anticipated for the proteome fraction of cell division protein X according to the threshold initiation model; exemplary proteins are presented in Fig EV4. It is of course also possible that the model is simply wrong, or that the proposed proteins X were not detected in the existing mass spec study, which is biased to detect highly expressed cytoplasmic proteins.

Finally, we note that recent results of single-cell studies (Campos *et al*, 2014; Taheri-Araghi *et al*, 2015) further constrain possible models of cell size control. For example, the results of Taheri-Araghi *et al* (2015) suggest that in order for the threshold initiator model to work, the division proteins X must be completely consumed at cell division, so that the cellular abundance of X reflects the amount of newly synthesized proteins. However, at the mean-field level, relevant to population-averaged, steady-state properties, these different single-cell models are equivalent. For example, the model that requires the addition of a fixed amount of protein X for cell division (due to the consumption of X in the division process) differs from the simpler threshold abundance model introduced here by a simple rescaling factor of the threshold $P_X^*$ when considering culture-averaged, steady-state properties. For this reason, we do not explicitly differentiate between various single-cell rules for division.

---

**Box 2: Growth rate dependence of cellular DNA content**

WT cells under nutrient limitation exhibit two distinct regimes according to the Helmstetter–Cooper (HC) model of bacterial chromosome replication (Appendix Fig S9): In the fast growth regime (doubling time DT < single-chromosome replication time, the "C-period"), the C-period is constant (at its minimal value) and the total DNA synthesis rate is determined by the replication initiation rate. In the slow growth regime (DT > C-period), chromosome replication is limited by the replication fork elongation rate, which is in turn limited by the synthesis of nucleotides (DNA monomers) (Neidhart, 1996). Under LacZ OE, the DNA content increases (Figs 1F and EV3A and B). Since multiple chromosome equivalents per cell are observed in a single nucleoid complex (Fig EV3), the HC model of DNA replication may still be applicable with multiple replication forks per cell, provided that the C-period > DT. The increase in DT under LacZ OE then implies that the C-period would have to increase at least as fast. This would present an interesting new regime for the coordination of DNA replication and cell growth, with the simultaneous occurrence of multiple rounds of DNA replication initiation and very slow rate of nucleotide synthesis.

Some insights into the slowdown of nucleotide synthesis, and the increase in average cellular DNA content under LacZ OE, can be gained from the perspective of proteome allocation. According to a recent proteomics study (Hui *et al*, 2015), the fraction abundance (per total proteome) of enzymes driving nucleotide synthesis, $\phi_{nuc}(\lambda)$, decreases linearly with the GR under LacZ OE, that is, $\phi_{nuc}(\lambda) \propto \lambda$ (see Appendix Fig S4). Assuming that this puts chromosome replication in the nucleotide-limited regime, then the rate of nucleotide synthesis, $\lambda \cdot \langle D \rangle$ where $\langle D \rangle$ denotes the average amount of DNA per cell, is proportional to the cellular abundance of enzymes of the nucleotide production pathways, $\phi_{nuc} \cdot \langle P \rangle$, that is, $\lambda \cdot \langle D \rangle \propto \phi_{nuc} \cdot \langle P \rangle$. This leads to a constant ratio of cellular DNA and protein content, $\langle D \rangle / \langle P \rangle \propto \phi_{nuc}/\lambda \propto$ const. Thus, as total protein per cell increases due to LacZ OE (Appendix Fig S7), an accompanying increase in DNA per cell would be expected based on this simple consideration.

However, while we did observe a several-fold increase in cellular DNA content (Fig 1F), at a quantitative level, this increase is smaller than expected from the above nucleotide-limited picture. In particular, the DNA–protein ratio $\langle D \rangle / \langle P \rangle$, often taken to be invariant under different conditions (Mortimer, 1958; Neumann & Nurse, 2007; Turner *et al*, 2012), decreased more than 2-fold at the slowest growth, for those cells under LacZ OE compared to those subjected to nutrient limitation (Appendix Fig S10A, compare red diamond and green circles, respectively). Possibly, LacZ OE affects the relative abundance of initiation factors such as DnaA or Ssb to some degree, resulting in lower rates of chromosomal initiations per protein. In any case, the LacZ OE system provides an interesting new window to investigate the coordination of DNA replication with cell growth in a non-classical regime.

---

8-fold overall difference between nutrient limitation and LacZ OE at the slowest GR. *A priori*, one might expect LacZ OE to result in an increased dry mass density and molecular crowding. Instead, a tight correlation was found between dry mass (also cellular protein) and physical cell size under all tested growth limitations (Figs 1G and EV1A), despite large changes in cell size (Fig 1C) and macromolecular composition (Appendix Fig S8). Correspondingly, the GR dependence of cell number (per volume of culture at constant $OD_{600}$), as determined by Coulter counter and colony count (Fig EV2), followed just the opposite trends as cell size. While most of the increase in cellular dry mass under LacZ OE was attributable to an increase in cellular protein (Appendix Fig S7), cellular RNA also exhibited a significant increase (Fig 1E).

The GR dependence of cellular DNA content (Fig 1F), confirmed by DAPI staining (Fig EV3A and B), exhibited similar trends as those of cell size (Fig 1C) with a strong correlation ($R^2 = 0.93$, see Fig EV1C). Compared to the well-known positive GR dependence of the cellular DNA content under nutrient limitation (Helmstetter & Cooper, 1968; Hill *et al*, 2012), growth limitation by LacZ OE again gave the opposite trend, with DNA content reaching a 4-fold higher level than that at similar GR under nutrient limitation. The measured DNA content, < 2 genome equivalent (DNA abundance in units of full chromosomes) under nutrient-limited growth as is well established (Helmstetter & Cooper, 1968), reached nearly eight genome equivalent at similar GR under LacZ OE (right vertical axis, Fig 1F). Microscopy images of DAPI-stained cells show that cell division proceeds normally, unlike cells resulting from the inhibition of cell division (Zaritsky *et al*, 2006).

## Discussion

The growth rate dependences of cell mass that we observe under the different perturbations demonstrate an important role of protein synthesis in the regulation of cell division and support a simple class of models of cell size control (Fig 2A, Box 1). In such a model, the initiation of cell division requires the cellular abundance of specific cell division proteins to reach a threshold level (Fantes *et al*, 1975; Wold *et al*, 1994; Donachie & Blakely, 2003). Figure 2A gives a schematic illustration of how this model leads to inflated cells under LacZ OE: As established in Hui *et al* (2015), under LacZ OE the proteome fraction of most proteins is reduced in a manner directly proportional to the reduced GR (see Appendix Fig S4). If cell division proteins follow this common trend, this model predicts a linearly divergent increase in total cell protein content $\langle P \rangle$ with decreasing GR (Box 1). Indeed, plotting the $1/\langle P \rangle$ vs. the GR (Fig 2B) reveals almost perfect direct proportionalities for both LacZ OE series (full and empty diamonds). Moreover, this model predicts that the fractional abundance of protein X (per total cellular protein) should generally follow the growth rate dependence of $1/\langle P \rangle$. The latter is presented in Fig 2B; the fractional abundances of some exemplary proteins matching the profile of Fig 2B are shown in Fig EV4. They are candidates of proteins X according to the model. Finally, we remark that threshold abundances of cell division proteins should be considered as a necessary, but not sufficient, condition for cell division to occur. Other cell division "checkpoints", like chromosomal replication or cellular elongation, may be required in addition to the threshold initiator checkpoint for cell division to proceed through completion.

Our study also revealed profound changes in cellular DNA content under different growth limitations, presented in Figs 1F and EV3A and B, posing the question to what extent the classical Helmstetter–Cooper model of bacterial chromosome replication (illustrated in Appendix Fig S9) holds under these conditions. The observed increase in cellular DNA content under LacZ OE can be rationalized from the perspective of proteome allocation in a nucleotide-limited regime, where a corresponding increase in cellular DNA accompanying the increase in cellular protein would be expected (Box 2). However, at a quantitative level, the observed increase in cellular DNA was smaller than expected, as the DNA–protein ratio,

often taken to be invariant under different conditions (Mortimer, 1958; Neumann & Nurse, 2007; Turner *et al*, 2012), exhibited more than 2-fold differences between growth limitations at slow growth rates (Appendix Fig S10A, compare red diamond and green circles, respectively), suggesting additional limitations of DNA synthesis under LacZ OE.

Finally, we remark on the tight correlation found between the average cell volume and dry mass (or protein content), across all modes of growth limitations studied here (Figs 1G and EV1A). This is well known for cells grown under nutrient limitation, as the cell's buoyant density was shown to change little under nutrient variation (Nanninga & Woldringh, 1985). Here, we find the same to hold for Cm inhibition and for the inflated cells produced by LacZ OE (Figs 1G and EV1A). *A priori*, one may have expected LacZ OE cells to exhibit a higher dry mass density like *E. coli* during steady-state growth in hyperosmotic conditions (Cayley & Record, 2004) and become more densely packed with protein leading to molecular crowding (Vazquez *et al*, 2008). Instead, the cell keeps nearly a constant ratio between dry mass and cell size, even when artificially forced to produce large quantities of "useless" protein. This coordination of the water and mass content could be mediated by mechanisms of osmoregulation, as for example, illustrated in the chemiosmotic model of Fig EV5: Coupling of partial charges of proteins and RNAs to intracellular osmolytes due to the Gibbs–Donnan effect (Donnan, 1911) affects osmotic pressure balance and thereby modulates cell size. Such a mechanism would naturally correct for fluctuations in the cell's buoyant density, as well as coordinate biomass production with cell volume growth in general.

# Materials and Methods

### Strains

The strains used in this study are either wild-type *E. coli* K-12 NCM3722 (Soupene *et al*, 2003; Lyons *et al*, 2011) or the LacZ overexpression strain NQ1389 described in Hui *et al* (2015).

### Growth media

All the minimal media are MOPS-buffered media described in Cayley *et al* (1989), which contains 40 mM MOPS and 4 mM tricine (adjust to pH 7.4 with NaOH), 0.1 mM $FeSO_4$, 0.276 mM $Na_2SO_4$, 0.5 μM $CaCl_2$, 0.523 mM $MgCl_2$, 10 mM $NH_4Cl$, 0.1 M NaCl, and also micronutrients used in Neidhardt *et al* (1974). Various carbons are used as specified below: 0.2% (w/v) glucose, 0.2% (v/v) glycerol, 60 mM sodium acetate, and 0.2% (w/v) mannose. In addition, rich defined medium (RDM) + glucose medium also contains 0.2% (w/v) glucose, micronutrients, various amino acids, nucleotides, and vitamins as described in Neidhardt *et al* (1974). Glucose + cAA medium contains 0.2% (w/v) glucose and 0.2% (w/v) casamino acids.

### Cell growth

Cell growth is performed with a 37°C water bath shaker (220 rpm). For the growth of NCM3722 strains, cells from a fresh colony in a LB plate were inoculated into LB broth and grown for several hours at 37°C as seed cultures. Seed cultures were then transferred into MOPS medium and grown overnight at 37°C as pre-cultures. Overnight pre-cultures were diluted to $OD_{600}$ around 0.01 to 0.02 in the same MOPS medium and grown at 37°C as experimental cultures. For the growth of NQ1389 (LacZ overexpression) strain with various levels of chlortetracycline (cTc), seed cultures and pre-cultures were not supplemented with cTc, and experimental cultures were first grown to an $OD_{600}$ of ~0.05 without cTc. Various concentrations of cTc were then added to the cultures, and cultures were grown for about three generations until reaching a new steady-state exponential phase.

### Total protein quantification

Total protein quantification method is the same as used by You *et al* (2013).

### Total RNA quantification

Total RNA quantification method is the same as used by You *et al* (2013).

### Total DNA quantification

Total DNA quantification is based on the diphenylamine colorimetric methods used by Bipatnath *et al* (1998) with modifications. Briefly, 10 ml of cell cultures in exponential phase ($OD_{600}$ = 0.3–0.5) was collected by centrifugation and immediately frozen in dry ice. Cell pellets were first washed once with 1 ml 0.45 M $HClO_4$ and then washed again with 1 mM $HClO_4$. The cell pellet was then hydrolyzed by 0.5 ml 1.6 M $HClO_4$ at 70°C for 30 min. After cooling to room temperature, 1 ml diphenylamine reagent (0.5 g diphenylamine in 50 ml glacial acetate, 0.5 ml 98% $H_2SO_4$, and 0.125 ml 32 mg/ml acetaldehyde water solution) was added for colorimetric reaction. After a 16-h to 18-h overnight reaction, the reaction mixture was centrifuged, and the $A_{600}$ of the supernatant was measured. At the same time, a series of standard calf thymus DNA (10 mg/ml in stock solution) reaction was set up in parallel, in order to obtain the DNA standard curve. Bacterial total DNA content was determined from the calf thymus standard curve.

### Dry weight measurement

About 250 ml of cell culture in exponential phase ($OD_{600}$ = 0.3–0.5) was collected by centrifugation. Cell pellets were resuspended in 200 ml $ddH_2O$ and collected again by centrifugation. Cell pellets were then suspended in 2 ml $ddH_2O$ and transferred to aluminium pans, and baked overnight until reaching constant weight. This weight corresponds to the dry weight.

### Bacteria cell counting

Bacterial cell counting was performed with a Multisizer 3 Coulter counter (Beckman Coulter). About 1 ml cell culture in exponential phase ($OD_{600}$ = 0.3–0.5) was collected by centrifugation. Cell pellets were washed once and finally dissolved in the 0.9% saline solution

(filtered by 0.22 μm Milipore Sericup filter). Cell samples were then filtered through a 11-μm nylon net filter (EMD Milipore) to remove large aggregates. Before the measurement, cell samples were further diluted 500-fold in 10 ml 0.9% saline solution. Cell counting was performed with a 20-μm aperture tube. Data were analyzed with MS-Multisizer 3 software (Beckman Coulter).

### Cell plating

For cell counting with plating, cell culture in exponential phase ($OD_{600}$ = 0.3–0.5) was serially diluted $10^6$-fold with the same growth medium (pre-warmed to 37°C). About 0.1 ml of the diluted cell sample was added to an LB plate. For each plate, eight small beads were added. The plate was then quickly shaken to uniformly spread the cell sample around the plate. The plate was further dried in a 37°C incubator for 1 h before removing the beads. Cells were grown for roughly 12 h at 37°C before counting the colonies. A typical plate has 50–200 colonies.

### General microscopy methods

About 8 μl of cell culture at $OD_{600}$ = 0.3–0.5 was applied to a cover slip, and covered with a 2-mm-thick layer of 2% agar in order to immobilize the cells and hold them flat to the cover slip. Cells were imaged using a 60× phase contrast objective (NA 1.40) with a Nikon Eclipse Ti-U inverted microscope. Images were obtained with a Clara (Andor) CCD camera. All image analysis was performed using the ImageJ suite of tools.

### Cell size measurement

Phase contrast images were captured immediately after sampling from exponential phase culture ($OD_{600}$ = 0.3–0.5) (see General microscopy methods). To obtain the cell size, individual cells were first identified by thresholding the image intensity such that the entire cell was selected, but none of the background. Next, the "Feret's diameter" was calculated for each cell, giving both the longest (length, $L$) and shortest (width, $W$) caliper distance along the boundary of the selected area. Cell volume ($V$) was calculated according to the equation $V = \pi R^2 \cdot (L - 2R/3)$. For each condition, between 500 and 1,000 individual cells were analyzed.

### DAPI staining

Our protocol closely follows Bernander *et al* (1998). Cell culture was grown for at least five generations to exponential phase ($OD_{600}$ = 0.3–0.5). About 1 ml culture was collected by centrifugation, washed twice in 1 ml ice-cold TE buffer (10 mM Tris–HCl, pH 8.0, 1 mM EDTA), and finally resuspended in 0.1 ml ice-cold TE buffer. Cells were fixed by adding 1 ml 77% ethanol. The fixed cell sample can be stored in 4°C. About 0.5 ml of fixed cell sample was collected by centrifugation and washed once with 1 ml Tris-$MgCl_2$ buffer (10 mM Tris–HCl, pH 7.4, 10 mM $MgCl_2$), and then resuspended in 250 μl 10 mM Tris-$MgCl_2$ containing 2 μg/ml DAPI (10 mg/ml stock). DAPI staining lasted for 15 min at room temperature. Stained cells were then imaged using excitation/emission filters at 360 nm (BW 40 nm) and 460 nm (BW 50 nm) (see also General microscopy methods). After background subtraction, the fluorescence intensity was integrated over the entire area of the cell to get the total DNA in each cell ("Integrated density" measurement in ImageJ). For every growth condition, 10–20 DAPI-stained cells were analyzed. Bars in Fig EV3 are the standard deviation of the integrated DAPI fluorescence of all measured cells.

**Expanded View** for this article is available online.

### Acknowledgements
We are grateful to Minsu Kim, Chenli Liu, Hugo Stocker, Suckjoon Jun, Marco Cosentino-Lagomarsino, and Matthew Scott, as well as Uwe Sauer and his laboratory for useful discussions, and to Tony Hui, Jessica V. Nguyen, and Rick A. Reynolds for technical assistance. This work was supported by SystemsX.ch (TPdFS) to MB, and by the NIH (R01-GM109069) and the Simons Foundation (Grant 330378) to TH. Manlu Zhu acknowledges the financial support from the China Scholarship Council (CSC, 201306010039).

### Author contributions
MB and TH designed the study. MB, MZ, XD, MW, and DS performed experiments. MB, MZ, XD, MW, Y-PW, and TH analyzed the data. MB, MZ, and TH wrote the paper and the supplement.

### Conflict of interest
The authors declare that they have no conflict of interest.

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
