## [Review Process File · Molecular Systems Biology]

Inflating bacterial cells by increased protein synthesis

Markus Basan, Manlu Zhu, Xiongfeng Dai, Mya Warren, Daniel Charles Sévin, Yi-Ping Wang and Terence Hwa

Corresponding author: Terence Hwa, University of California at San Diego

Review timeline:

Submission date:	19 March 2015
Editorial Decision:	04 May 2015
Revision received:	02 August 2015
Editorial Decision:	10 September 2015
Revision received:	27 September 2015
Accepted:	30 September 2015

Editor: Maria Polychronidou

Transaction Report:

1st Editorial Decision

04 May 2015

Thank you again for submitting your work to Molecular Systems Biology. We have now heard back from the three referees who agreed to evaluate your manuscript. Overall, the referees acknowledge that the presented findings seem potentially interesting. However, they raise a series of concerns, which should be carefully addressed in a revision of the manuscript.

As you will see from the reports below, reviewers #1 and #2 are cautiously positive and mostly raise relatively minor issues (i.e. related to text modification and providing clarifications throughout the manuscript). However, reviewer #3 is more critical and refers to the need to apply the presented observations for deriving new conclusions (i.e. in the context of the "threshold initiation model of bacterial cell-size control"), in order to enhance the overall impact of the study.

In line with the comments of the reviewers, we think that the text should be carefully re-written in order to make the main findings/conclusions more easily accessible to the reader. Moreover, some of the key experimental data should be moved from the Supplement into the main figures.

If you feel you can satisfactorily deal with these points and those listed by the referees, you may wish to submit a revised version of your manuscript. Please attach a covering letter giving details of the way in which you have handled each of the points raised by the referees. A revised manuscript will be once again subject to review and you probably understand that we can give you no guarantee at this stage that the eventual outcome will be favorable.

Reviewer #1:

the paper reports what I find to be very interesting results: that over-expression of 'unneeded' proteins result in large increase in cell mass, proteins, RNA and DNA content. These over-expressing cells decrease their specific growth rate, as shown before. The increase in these different physiological parameters with decreasing growth rate therefore contrasts the inverse correlation typically observed when cells are grown in different media (and that are shown again here).

The results have clear implications for understanding how cells coordinate protein production, growth rate and size and should therefore be of wide interest.

I'm much less enthusiastic about the way the paper was put together and I must say that I found it highly confusing and difficult to follow. First, from a technical point of view, the paper has one result figure in main text, and about ten supplementary figures. I don't see how this can be justified. The relative weight of the (narrow) result section and the (very long) discussion is also quite unusually. While this could perhaps be justified in certain cases, I found the discussion section not organized and quite confusing, trying to address too many in semi-rigorous manner, but in fact doesn't follow up completely on any of the points addressed. There are too many assumptions required for the different points mentioned; these are not explicitly defined, and the consequences not explicitly derived. The introduction also reads as if it had been put through too quickly - will be very difficult to follow for people non-expert.

with regards to the discussion - I think the authors have tried too hard to come up with a unified model that will explain all. Perhaps the times are too early for that? I would feel much more satisfied with a discussion that will raise the interesting questions that are raised by the new data, will point to different possibilities (and perhaps short-coming of previous models), and will highlight possible experimental directions to address that. Alternatively, if the authors do have some unified model that explains all already at this stage, they should be more explicit about exactly the assumptions, consider all aspects of their data (the observed increase in DNA, RNA, proteins - what does it mean on transcription/translation etc.?) be more explicit one how the cells adjust to these changes etc.

so in my opinion the paper requires really extensive rewriting and presentation.

I also didn't understand why the data about RNA increase is not shown and discussed in main text. The authors mention that increase is small, but I don't think the data shows that, in particular during intermediate change in growth rate (about two-folds reduction) where the changes are equivalent to those seen in the other parameters. This also has clear and highly important implications and should therefore be discussed.

if at the very slow growing cells the reduction is lower, the authors should relate to the point that RNA decreases less than the DNA content.

The distribution of sizes and the OE strains is very wide. Is it clear that this is indeed a steady state distribution that doesn't change in time? could it be that cells become larger and larger in time? what does the distribution look like as a function of OD?

there are no error-bars to the figures. what is the variability that is observed? what is the background / errors in the different parameters? this is particularly important for comparing changes in e.g. RNA levels to changes in different parameters and to understand the relative slow changes in some parameters in the 0.5-1.5 growth-rate regime.

no growth rate curves are shown. how long is the exponential phase? is growth indeed exponential under all conditions reported? any information about death-rate? is it becoming substantial? depends on cell size?

Reviewer #2:

In 'Inflating bacterial cells by protein synthesis' Basan et al consider a series of orthogonal growth conditions and measure various physiological properties. The most interesting one greatly overproduced a useless protein, and showed that this inverted many of the expected relationships between properties of cells. The kind of 'growth laws' and general relationships between physiological properties promoted by the Hwa group and others always struck me as highly interesting and important, and I quite liked this paper. I therefore recommend the manuscript for publication in MSB. That said, I would also like to see some improvements.

My main scientific concern is that I would have liked to see the over-expression done with at least one other protein, e.g. a dark FP or some other useless protein. The results are probably not unique to LacZ, but that is also hard to know without trying at least one other protein. I would not necessarily hold my recommendation ransom if this is not feasible, but it strikes me as a rather straightforward thing to do. Virtually all of my other comments are related to readability. The brevity with which the results are presented makes it quite hard to read the paper even if the ideas are not that complicated.

In the second paragraph of the introduction the authors mention a simple model of size control, in which the total amount of a regulatory protein needs to reach a fixed threshold abundance before division takes place. This is an idea I am rather closely familiar with, and I believe the growth laws observed by e.g. the Jun lab support a slightly different picture: it is not a threshold abundance that must be reached, but a threshold amount of de novo synthesis. The two would be the same only if the regulatory protein in question were consumed in the division process, which then should be stated in the text. For example, large newborn cells start out with more protein than small cells, and the observation that the total amount of cell mass added is independent of the starting point seems to go against models of a fixed total abundance unless this assumption is added. This needs to be considered, and the authors should make sure their conclusions are not affected by the difference in the two models.

In the same paragraph, and throughout, the definitions are a little hard to track. Is P defined anywhere? I imagine it is the total protein abundance and maybe it could be called P_{tot} ? And what does P^* mean? Presumably the total proteome abundance at the time of division, but the asterisk was defined mechanistically, as the threshold in a particular regulatory protein X that controls division. This makes little sense for the whole proteome, and I guess the authors mean the average value of the proteome at division? Perhaps this is a subtle/minor point but it made the text harder to read.

The last sentence on page 5 mentions that "It is of course also possible that the regulator X was not detected in the existing mass spec study, which is biased to detect highly expressed cytoplasmic proteins". On this occasion and many others, the authors seem to make an assumption earlier in the text, and later treat this almost as a known fact. For example, the notion that cells divide at a particular threshold concentration of some regulatory protein is controversial in itself, and the sentence about "the regulator X " seems to treat it as a given. It happens to be the explanation that I favor myself, but many other explanations have also been suggested. The discussion really needs to be more carefully written to separate what parts of the conclusions are based on speculative assumptions and what parts are based on measured facts.

In the bottom paragraph of page 6, there is a sentence stating "Assuming that this puts chromosome replication in the nucleotide-limited regime...". I am not sufficiently familiar with the literature to know if that assumption is supported by other findings, and maybe the authors could elaborate more. I was certainly surprised by the assumption, though it could be a reasonable one. This whole paragraph was also not easy to read.

Figure 1B could do with some edits/additions. First, when distributions are plotted in linear scale the ones with higher average look broader, even if they may not be broader relative to the average. I would therefore like to see the x-axis in log-scale, or at least an inset with either log-scale or with the cell volume normalized by the average cell volume. Also, I would have preferred to see the distributions compared for similar growth rates, which according to Fig. 1C the authors should have access to. As it stands now, Fig. 1B seems to make a statement about the distributions corresponding

to the different growth conditions (sugar, OE, drug etc) but judging from Fig. 1C it seems that similar differences in fact could be observed within each such condition. This is perhaps also a place where more analysis could be useful in general. To what extent do the distributions differ from the ones expected for exponentially growing cells (published several times in the 1950s and 1960s and then again, I believe, in the Helmstetter and Cooper paper cited)? How similar are the distributions when comparing for the same growth rate, or for the same average size? If the distributions are judged important enough to include in the paper, then surely it must be worth evaluating them enough to draw some sort of conclusion from them. Now they are just said to be 'distinct', but since they are for different growth rates, that an empty statement.

Minor points:

The term 'equivalent to' in the abstract is a little confusing. The term 'remarkably' in the next sentence seems a little strong since this struck me as the least remarkable feature of the paper.

The tight connection between dry mass and cell size is argued to suggest "a mechanism of coordination that is both precise and robust". By using words such as coordination, precise and robust, the authors, perhaps inadvertently, paint a picture of an active process regulating multiple things in order to achieve this effect, when the correlation almost seems hard to avoid. I suggest either elaborating or rewriting.

The last statement, that the approach may be applicable to other organisms, seemed like a weak ending. Perhaps the authors could come up with a more exciting outlook paragraph that is also more connected to what they find.

Typos:

A right-parenthesis is missing in reference to figure S7C

Reviewer #3:

This study examines bacterial cell size under various growth conditions. The paper is centered around the observation that Cm treatment and LacZ OE deviate from the well-known cell-size dependence on growth rate. While this finding is interesting and I appreciate the authors' coarse-grain approach and various measurements of macromolecules, I feel that the paper does not go beyond this observation. The authors discuss several aspects of E. coli physiology (cell size regulation, coordination of DNA replication and cell growth, and tight correlation between cell volume and dry mass), but no concrete conclusion can be drawn. For example, the study neither supports or disproves the 'threshold initiation model' of bacterial cell-size control. The authors' discussion offers only an interpretation of their observation in the context of the model. I also find their interpretation questionable (see below).

<Major comments>

- About the threshold initiation model. The authors state that plotting the inverse of cellular mass vs GR can reveal the GR dependence of, ϕ_X (i.e. P_X/P). This implicitly assumes that P_X itself is independent on GR, which is not supported. In fact, multiplying Fig. S3D (P_X/P) by Fig. S6A (P) appears to suggest that P_X (or proteome sector C, A, U or S, to which the regulator X presumably belong), be dependent on GR.

- Related to the comment above, what exactly is P^* (characteristic total protein content of the cell)? Because P^*_X is defined as a threshold abundance of X to trigger cell division, I initially thought it was the per-cell total protein at cell division (or when cell division is triggered). However, what's shown in Fig. 2B, which is referred to as $1/P^*$ vs GR, is $1/[\text{mean per-cell total protein}]$ vs GR.

- On pg. 6, the authors state that "while we did observe a several-fold increase of cellular DNA content (Fig. 1F), at a quantitative level, this increase is smaller than expected from the above nucleotide-limited picture". What's the basis of this statement? Is it simply because in Fig. S13A, the D/P ratio decreased with decreasing GR rather than staying flat? If so, it could also be caused by more-than-expected increase in protein mass. Therefore, I don't find their statement, "our results

suggest that under LacZ OE, DNA synthesis is not limited by nucleotide availability, but by other factors involving perhaps replication or elongation", to be supported adequately.

|
<Minor comments>

- It would be more informative to have per-cell values in Fig. S8
- On pg. 6, 2nd paragraph, define 'D' earlier.
- In Fig. S3B, what are different symbols?

Reviewer #1:

the paper reports what I find to be very interesting results: that over-expression of 'unneeded' proteins result in large increase in cell mass, proteins, RNA and DNA content. These over-expressing cells decrease their specific growth rate, as shown before. The increase in these different physiological parameters with decreasing growth rate therefore contrasts the inverse correlation typically observed when cells are grown in different media (and that are shown again here).

The results have clear implications for understanding how cells coordinate protein production, growth rate and size and should therefore be of wide interest. I'm much less enthusiastic about the way the paper was put together and I must say that I found it highly confusing and difficult to follow.

We would like to thank the reviewer for the overall positive evaluation. As described below, we have made a big effort in rewriting our manuscript to address the concerns raised, resulting in improved clarity of the presentation.

First, from a technical point of view, the paper has one result figure in main text, and about ten supplementary figures. I don't see how this can be justified.

We would like to point out that this manuscript was submitted as an MSB short report, which we believe is the appropriate format for our results. Our finding is simple – we obtained data on protein, RNA, DNA, cell size. All original data are presented in Figure 1, which is now expanded. In the majority of our supplemental figures containing data, they do not constitute additional data, but rather different ratios and different analysis of data already presented in the main text. We believe that such analysis is valuable to some readers, but may be too distracting in the main text. We will leave it to the editor to decide on the preferred format.

The relative weight of the (narrow) result section and the (very long) discussion is also quite unusually. While this could perhaps be justified in certain cases, I found the discussion section not organized and quite confusing, trying to address too many in semi-rigorous manner, but in fact doesn't follow up completely on any of the points addressed. There are too many assumptions required for the different points mentioned; these are not explicitly defined, and the consequences not explicitly derived. The introduction also reads as if it had been put through too quickly - will be very difficult to follow for people non-expert.

with regards to the discussion - I think the authors have tried too hard to come up with a unified model that will explain all. Perhaps the times are too early for that? I would feel much more satisfied with a discussion that will raise the interesting questions that are raised by the new data, will point to different possibilities (and perhaps short-coming of previous models), and will highlight possible experimental directions to address that. Alternatively, if the authors do have some unified model that explains all already at this stage, they should be more explicit about exactly the assumptions, consider all aspects of their data (the observed increase in DNA, RNA, proteins - what does it mean on transcription/translation etc.?) be more explicit one how the cells adjust to these changes etc.

so in my opinion the paper requires really extensive rewriting and presentation.

We are aware of the problem the reviewer is raising regarding the Discussion section and have extensively rewritten the manuscript to clarify the presentation, which we acknowledge was not optimal.

The aim of our Discussion section is not to come up with a unified model. We agree with the reviewer that the time is not ripe for an all-encompassing model of cell size and cellular composition. We simply want to put our findings in a broader perspective and perform a quantitative analysis of the conclusion that can be drawn from it. We believe that this type of analysis is valuable and can help provide a basis for future work.

We have therefore extensively rewritten our manuscript to address the reviewer's concerns.

In particular,

- We have completely rewritten the introduction section.
- We have significantly shortened and rewritten the discussion section.
- We have moved all mathematical details of the models to two Supplementary Notes.
- We have extensively rewritten and changed the order of presentation for the threshold initiator model, which is now one of the Supplementary Notes.
- We have added a cautionary comments discussing the limitations of the threshold initiator interpretation and also commented on the relation of our mean-field treatment to single-cell models in the Supplementary Note.

- We have rewritten the discussion about the increase in DNA level, and moved the details to the other Supplementary Note.

I also didn't understand why the data about RNA increase is not shown and discuss in main text. the authors mention that increase is small, but I don't think the data shows that, in particular during intermediate change in growth rate (about two-folds reduction) where the changes are equivalent to these seen in the other parameters. This also has clear and highly important implications and should therefore be discussed.

if at the very slow growing cells the reduction is lower, the authors should relate to the point that RNA decreases less than the DNA content.

We have included the response of cellular RNA in the expanded Fig.1 as the reviewer requested, and we now discuss this result in our revised manuscript. However, we do point out that the increase in RNA is proportionally smaller than the increase in cellular protein and even DNA. The majority of cellular RNA is attributable to ribosomal RNA (Scott, Gunderson, Mateescu, Zhang, & Hwa, 2010), which is why we mainly consider RNA as a proxy for cellular ribosome content.

The distribution of sizes and the OE strains is very wide. Is it clear that this is indeed a steady state distribution that doesn't change in time? could it be that cells become larger and larger in time? what does the distribution look like as a function of OD?

We thank the reviewer for requesting this important data that will make our findings more convincing. This is now included in our revised manuscript, as Figs. 2E, F comparing normalized distributions at different OD₆₀₀ for different levels of LacZ OE and in Table S3 for a summary of additional measurements at different OD₆₀₀. As can be seen from these results, cell size distributions are highly reproducible and independent of OD₆₀₀.

All measurements were taken from exponentially growing cell cultures and we have now included growth curves for all conditions in our revised manuscript (Fig. S1). Cell division appears to proceed normally in large LacZ OE cells (see also Fig. S11C for microscopy images).

there are no error-bars to the figures. what is the variability that is observed? what is the background / errors in the different parameters? this is particularly important for comparing changes in e.g. RNA levels to changes in different parameters and to understand the relative slow changes in some parameters in the 0.5-1.5 growth-rate regime.

We have included a new table (Table S3) presenting the results of cell size measurements from multiple biological repeats and measurements at different culture densities OD₆₀₀. These results demonstrate high reproducibility and independence of OD₆₀₀. Error bars for all the other measured quantities are given in Table S4.

The results presented in Tables S3-S4 demonstrate that the error bars are in general very small compared to the magnitude of the differences between different growth limitations and also the magnitude of the changes observed for individual limitations.

no growth rate curves are shown. how long is the exponential phase? is growth indeed exponential under all conditions reported? any information about death-rate? is it becoming substantial? depends on cell size?

As pointed out above, all measurements were taken from exponentially growing cell cultures and we have now included growth curves for all conditions in our revised manuscript; see Fig. S1. To minimize the occurrence of spontaneous mutations, we followed a standard procedure used for protein overexpression. Cells were first grown exponentially to OD 0.05 before LacZ OE was induced. Cells were then given 2-3 doublings to reach their new steady-state exponential growth rate, as presented in Fig. S1.

The plating results presented in Fig. S9 demonstrate that cell death does not play a significant role in the LacZ OE strain. There appears to be a small, approximately constant viability effect for chloramphenicol limitation.

Reviewer #2:

In 'Inflating bacterial cells by protein synthesis' Basan et al consider a series of orthogonal growth conditions and measure various physiological properties. The most interesting one greatly overproduced a useless protein, and showed that this inverted many of the expected relationships between properties of cells. The kind of 'growth laws' and general relationships between physiological properties promoted by the Hwa group and others always struck me as highly interesting and important, and I quite liked this paper. I therefore recommend the manuscript for publication in MSB. That said, I would also like to see some improvements.

We thank the reviewer for his/her general support, and for pointing out important issues to improve the clarity of our manuscript. We have substantially rewritten our manuscript in response to addressing the reviewer's comments.

My main scientific concern is that I would have liked to see the over-expression done with at least one other protein, e.g. a dark FP or some other useless protein. The results are probably not unique to LacZ, but that is also hard to know without trying at least one other protein. I would not necessarily hold my recommendation ransom if this is not feasible, but it strikes me as a rather straightforward thing to do.

One point where we differ from the reviewer is to repeat our study with a different OE protein. We believe that repeating the study using another protein is not essential (as acknowledged by the reviewer). There is substantial evidence from past studies (Scott et al., 2010), suggesting a highly comparable cellular response to the overexpression of different proteins: the absolute amplitude of growth reduction by LacZ OE collapsed with results reported by other labs using a variety of other systems.

Unfortunately, these experiments are not as easy as they might appear. Replacing the lacZ gene by, e.g., gfp using the same construct is of course easy. But it does not yield nearly as much gene expression (hence not much growth defect). This can be due to a number of reasons: translation efficiency may differ, and the gene length itself can play a big role. LacZ is 4x times longer than gfp. The alternative is to change the expression system, which would take some trials to get OE to be just in a good range for this kind of study. By the way, quantifying the absolute abundance of the OE protein would also generally be a problem, if it is to be done repeatedly until a good range is found. We are concerned that further work along this line would result in a significant delay and may result in the publication of our main findings by other groups, who are aware of our results and are pursuing similar directions.

Virtually all of my other comments are related to readability. The brevity with which the results are presented makes it quite hard to read the paper even if the ideas are not that complicated.

In the second paragraph of the introduction the authors mention a simple model of size control, in which the total amount of a regulatory protein needs to reach a fixed threshold abundance before division takes place. This is an idea I am rather closely familiar with, and I believe the growth laws observed by e.g. the Jun lab support a slightly different picture: it is not a threshold abundance that must be reached, but a threshold amount of de novo synthesis. The two would be the same only if the regulatory protein in question were consumed in the division process, which then should be stated in the text. For example, large newborn cells start out with more protein than small cells, and the observation that the total amount of cell mass added is independent of the starting point seems to go against models of a fixed total abundance unless this assumption is added. This needs to be considered, and the authors should make sure their conclusions are not affected by the difference in the two models.

We thank the reviewer for pointing out this subtle issue, which we had forgotten to address in our initial submission.

These two models are of course very different, when considering single cell dynamics. However, in our manuscript, we only consider steady-state growth conditions that are averaged over the cell population. For these purposes, the difference whether or not critical cell division proteins are consumed in the division process (meaning that addition is what is required), does not affect our conclusions. The mean-field version of a model with the consumption of cell division proteins in the division process, is equivalent to our model up to a factor 2 in the required threshold abundance.

Therefore, we do not explicitly take these subtle differences in the underlying single-cell division rules into account. However, we have reformulated the presentation of the threshold initiator model in the light of the reviewer's comments. We now explicitly discuss this point in Supplementary Note 1.

In the same paragraph, and throughout, the definitions are a little hard to track. Is P defined anywhere? I imagine it is the total protein abundance and maybe it could be called P_tot? And what does P* mean? Presumably the total proteome abundance at the time of division, but the asterisk was defined mechanistically, as the threshold in a particular regulatory protein X that controls division. This makes little sense for the whole proteome, and I guess the authors mean the average value of the proteome at division? Perhaps this is a subtle/minor point but it made the text harder to read.

We thank the reviewer for pointing out these problems in our presentation of the model. We realize that our previous presentation was the cause of some confusion. We have substantially rewritten the main text to clarify these issues. In particular, we have modified the notation used in the model and now explicitly differentiate between single cell properties, e.g. P, and properties averaged over the population, which we now denote by $\langle \dots \rangle$, e.g. $\langle P \rangle$.

The last sentence on page 5 mentions that "It is of course also possible that the regulator X was not detected in the existing mass spec study, which is biased to detect highly expressed cytoplasmic proteins". On this occasion and many others, the authors seem to make an assumption earlier in the text, and later treat this almost as a known fact. For example, the notion that cells divide at a particular threshold concentration of some regulatory protein is controversial in itself, and the sentence about "the regulator X" seems to treat it as a given. It happens to be the explanation that I favor myself, but many other explanations have also been suggested. The discussion really needs to be more carefully written to separate what parts of the conclusions are based on speculative assumptions and what parts are based on measured facts.

We agree with the reviewer and have significantly rewritten the discussion in our manuscript. Details of the models have been moved to the Supplementary Material. The hypothetical nature of these ideas should now be clear in our revised manuscript.

In the bottom paragraph of page 6, there is a sentence stating "Assuming that this puts chromosome replication in the nucleotide-limited regime...". I am not sufficiently familiar with the literature to know if that assumption is supported by other findings, and maybe the authors could elaborate more. I was certainly surprised by the assumption, though it could be a reasonable one. This whole paragraph was also not easy to read.

We consider the *ad hoc* assumption of nucleotide-limited DNA synthesis, simply because this is the simplest scenario. However, this assumption is consistent with proteomics data that demonstrates that nucleotide synthesis pathways are compressed uniformly under LacZ OE (Hui et al., 2015). Indeed, the strong increase in cellular DNA can be rationalized using this simple assumption. However, as we point out in the text, quantitatively, the increase in cellular DNA is smaller than expected from this consideration, suggesting a role of other limiting factors like elongation or initiation.

Evidence for the assumption of nucleotide-limited DNA synthesis in the low growth rate regime of nutrient limited growth comes from metabolomics results which will be presented elsewhere.

We have rewritten the presentation of this paragraph and moved many details to the Supplementary Note 2 to simplify the presentation.

Figure 1B could do with some edits/additions. First, when distributions are plotted in linear scale the ones with higher average look broader, even if they may not be broader relative to the average. I would therefore like to see the x-axis in log-scale, or at least an inset with either log-scale or with the cell volume normalized by the average cell volume.

We follow the recommendation of the reviewer and in our revised manuscript, we have added the distributions normalized by average cell size as an inset in Fig. 1B. As the reviewer correctly anticipated, the distribution looked broader due to the linear scale. Normalizing the plots by the mean essentially collapsed them to the same size, in accordance with what the Jun lab found for cell size distribution under nutrient-limited growth. It was not our intention to suggest differences in the shape of the cell size distribution beyond the changes caused by a different mean.

For the convenience of the reviewer, we also attach the plot with the x-axis in log scale below.

Also, I would have preferred to see the distributions compared for similar growth rates, which according to Fig. 1C the authors should have access to.

This was precisely the idea of Fig. 1C. Except for the glucose condition, all distributions are at very similar growth rates. The glucose condition was included as a reference point. We have tried to clarify this point in our revised manuscript and this is now explicitly stated in the main text and the caption of Fig. 1.

As it stands now, Fig. 1B seems to make a statement about the distributions corresponding to the different growth conditions (sugar, OE, drug etc) but judging from Fig. 1C it seems that similar differences in fact could be observed within each such condition. This is perhaps also a place where more analysis could be useful in general. To what extent do the distributions differ from the ones expected for exponentially growing cells (published several times in the 1950s and 1960s and then again, I believe, in the Helmstetter and Cooper paper cited)? How similar are the distributions when comparing for the same growth rate, or for the same average size? If the distributions are judged important enough to include in the paper, then surely it must be worth evaluating them enough to draw some sort of conclusion from them. Now they are just said to be 'distinct', but since they are for different growth rates, that an empty statement.

As outlined above, it was never our intention to imply a special effect regarding the changes in shape of the distribution beyond the change in average cell size and thank the reviewer for pointing out this point, which has led to some confusion. We now show the distributions normalized by the mean as an inset of Fig. 1B, which shows that the different distributions are indeed very similar for the different limitations. We explicitly state in the caption of Fig. 1 that the different widths of the distribution are mainly caused by differences in the mean cell size. Cell size distribution is actually not essential to the main point of our work; we are happy to remove these distributions from the text if the reviewer feels this would be less confusing.

Minor points:

The term 'equivalent to' in the abstract is a little confusing.

We have modified the text to clarify this issue and replaced the second part of the sentence with "reaching up to an amount equivalent to ~8 chromosomes per cell."

The term 'remarkably' in the next sentence seems a little strong since this struck me as the least remarkable feature of the paper. The tight connection between dry mass and cell size is argued to suggest "a mechanism of coordination that is both precise and robust". By using words such as coordination, precise and robust, the authors, perhaps inadvertently, paint a picture of an active process regulating multiple

things in order to achieve this effect, when the correlation almost seems hard to avoid. I suggest either elaborating or rewriting.

We have removed the term 'remarkably' in the abstract and the qualitative statement "..., suggesting a mechanism of coordination that is both precise and robust". However, on the overall interpretation of this point, we respectfully disagree with the reviewer, as we do not believe that the correlation is trivial. Alternatively, one could have imagined that LacZ OE leads to cells with higher protein mass, but similar volumes due to a higher density of proteins. Such an increase in dry mass density is observed for example during steady-state growth in hyperosmotic condition (manuscript in preparation). Instead, cells manage to keep their dry mass density almost perfectly constant. Cells accomplish this feat, despite the fact that the overexpression of large amounts of LacZ is artificially induced. In other words, how do cells know about the large amount of additional useless protein present in their cytoplasm to adjust their volume accordingly? We have tried to clarify this issue in our revised manuscript in the discussion section.

The last statement, that the approach may be applicable to other organisms, seemed like a weak ending. Perhaps the authors could come up with a more exciting outlook paragraph that is also more connected to what they find.

We have tried to clarify the connection of our conclusion with our findings. Nevertheless, we believe one of the most exciting directions for future research is applying our approach to address cell size regulation in higher cells and tissues.

Typos:

A right-parenthesis is missing in reference to figure S7C

We have corrected this.

Reviewer #3:

This study examines bacterial cell size under various growth conditions. The paper is centered around the observation that Cm treatment and LacZ OE deviate from the well-known cell-size dependence on growth rate. While this finding is interesting and I appreciate the authors' coarse-grain approach and various measurements of macromolecules, I feel that the paper does not go beyond this observation.

We thank the reviewer for appreciating our finding as interesting and for the quantitative characterization of macromolecular composition. The main purpose of our submission is indeed to report these highly surprising and previously unreported observations. Due to the broad interest and implications of our findings (as also acknowledged by the other reviewers), we believe an "MSB Report" (which focus on reporting "a particularly provocative and novel aspect of a study") to be the appropriate format.

The authors discuss several aspects of E. coli physiology (cell size regulation, coordination of DNA replication and cell growth, and tight correlation between cell volume and dry mass), but no concrete conclusion can be drawn.

While we offer rationalizations of our findings in the context of models, we acknowledge that more work will be required to uncover the mechanisms underlying the multiple observations reported in our work. However, we believe that due to the diversity of the findings reported in our manuscript (ranging from cell size, DNA content to dry mass density) and the long-standing nature of the involved questions (cell size regulation), validating the underlying mechanisms goes far beyond the scope of our work, in particular considering our submission as a "Report" as defined by MSB.

For example, the study neither supports or disproves the 'threshold initiation model' of bacterial cell-size control. The authors' discussion offers only an interpretation of their observation in the context of the model. I also find their interpretation questionable (see below).

We regret that our initial submission was not written sufficiently clearly, leading to some misunderstandings as expressed above. We are grateful to the reviewer for pointing out his/her concerns. In fact, our observations can be naturally interpreted in the context of the threshold initiator model and therefore constitute (indirect) support of this model.

The inverse of average cellular protein $1/\langle P \rangle$ (in the improved notation) shows a remarkable, close to perfect direct proportionality to GR both in glucose minimal medium and glucose+cAA (Fig. 2B). This means that cell size diverges linearly as GR approaches zero. This is precisely the GR dependence of cell size expected from the threshold initiation model, as we explain now in Supplementary Note 1.

We have substantially rewritten our manuscript to improve its clarity and readability and to address the important issues raised by the reviewer, which we believe has resulted in a much improved manuscript.

<Major comments>

- About the threshold initiation model. The authors state that plotting the inverse of cellular mass vs GR can reveal the GR dependence of, ϕ_X (i.e. P_X/P). This implicitly assumes that P_X itself is independent on GR, which is not supported. In fact, multiplying Fig. S3D (P_X/P) by Fig. S6A (P) appears to suggest that P_X (or proteome sector C, A, U or S, to which the regulator X presumably belong), be dependent on GR.

We thank the reviewer for pointing out this issue which arose due to the lack of clarity in our presentation. In the threshold initiation model, the amount of the regulator protein per cell, P_X^* , is explicitly defined to be constant (the threshold value) which is independent of the growth rate. This means that $P_X^* = \phi_X \cdot \langle P^* \rangle = constant$, where $\langle P^* \rangle$ is the total protein abundance of the cell at the time of cell division (averaged over the cell population). Taking $\langle P^* \rangle$ to be related to the average protein abundance per cell $\langle P \rangle$ by a constant (due to asynchronous cell division throughout the population), we have $\phi_X \cdot \langle P \rangle = constant$. This leads to our expectation that plotting $1/\langle P \rangle$ vs GR yields the GR-dependence of ϕ_X . Thus, this comes directly from the definition of the threshold initiation model.

Our idea is to plot $1/\langle P \rangle$ vs GR (Fig. 2B) and compare to proteomic results (Hui et al, 2015) under different modes of growth limitation to see which proteins have abundance profiles (ϕ_X) similar to $1/\langle P \rangle$. Several such proteins are singled out in this way (as shown in Fig. S12) and are being further investigated. We believe this procedure itself is sound, up to the applicability of the threshold initiator model.

The reviewer pointed out a case of apparent contradiction: Under LacZ OE, $1/\langle P \rangle$ exhibits a directly proportional dependence on GR (red symbols in Fig. 2B). However, the abundances of the proteome sectors C, A, U, or S exhibit GR dependences which are approximately linear, but with vertical offsets (Fig. S4D, old Fig. S3D). The reviewer noted that if the regulator X does belong to one of these sectors, then it is different from the direct proportional difference exhibited by $1/\langle P \rangle$. This led him/her to conclude that the threshold initiator model is wrong.

While this deduction is natural, it is not justified based on some details of the proteome sector analysis in Hui et al that we neglected to describe in the original manuscript. The proteome-sector classification by Hui et al. was based on GR-dependence of the relative protein abundances under different modes of growth limitation. Proteins exhibiting similar trend in GR but with different vertical offsets are all included in the same sector. Thus, just because the total abundance of a proteome sector exhibits a linear GR-dependence with a non-zero offset, it does not mean every member of this sector has this vertical offset. Indeed, many individual proteins exhibit little vertical offset under LacZ OE; see Fig. S4C for some examples, with thorough analysis described in Hui et al. Thus there is no contradiction, and we believe that our way of using ϕ_X to profile the regulator X is sound.

Having been made aware of this problem by the reviewer's comment, we have provided a more thorough explanation of the rationale of this analysis and the difference between the abundance of a sector vs individual proteins in the caption of Fig. 4D.

We have also reorganized and rewritten the section discussing the threshold initiation model (now Supp Note 1) and improved the mathematical notation. We believe that this revised presentation clarifies this issue.

- Related to the comment above, what exactly is P^* (characteristic total protein content of the cell)? Because P^*_X is defined as a threshold abundance of X to trigger cell division, I initially thought it was the per-cell total protein at cell division (or when cell division is triggered). However, what's shown in Fig. 2B, which is referred to as $1/P^*$ vs GR, is $1/[\text{mean per-cell total protein}]$ vs GR.

Referring to Fig. 2B as $1/P^*$ in the main text directly was indeed misleading and we apologize for this confusion. In our revised manuscript, the average cellular protein at cell division is denoted by $\langle P^* \rangle$. $\langle P^* \rangle$ is proportional to the average cell size, but not identical. In our revised manuscript, we now point this out explicitly.

To summarize:

P^*_X is the amount of the regulator X to trigger cell division in a given cell. $\langle P^* \rangle$ is the total cellular protein at cell division averaged over the cell population. The average cellular protein in the population $\langle P \rangle$ is a proxy for and proportional to $\langle P^* \rangle$. For example in a homogeneously distributed population of cells at different times of progression throughout the cell cycle, where cellular protein varies between the size P^* at division and $P^*/2$ immediately after division, $\langle P \rangle$ would be given by $\frac{3}{4}\langle P^* \rangle$.

$1/\langle P \rangle$ exhibits a striking direct proportionality to growth rate, as expected from the threshold initiation model.

- On pg. 6, the authors state that "while we did observe a several-fold increase of cellular DNA content (Fig. 1F), at a quantitative level, this increase is smaller than expected from the above nucleotide-limited picture". What's the basis of this statement? Is it simply because in Fig. S13A, the D/P ratio decreased with decreasing GR rather than staying flat? If so, it could also be caused by more-than-expected increase in protein mass. Therefore, I don't find their statement, "our results suggest that under LacZ OE, DNA synthesis is not limited by nucleotide availability, but by other factors involving perhaps replication or elongation", to be supported adequately.

A previous proteomics study (Hui et al, 2015) shows that the relative abundance of nucleotide synthesis enzymes per protein (defined as $\phi_{nuc}(\lambda) = \langle P_{nuc} \rangle / \langle P \rangle$, where $\langle P_{nuc} \rangle$ is the average cellular amount of nucleotide synthesis pathways) is compressed linearly, $\phi_{nuc}(\lambda) \propto \lambda$, under LacZ OE. Therefore, if DNA synthesis were purely limited by nucleotides, we would have $\lambda \langle D \rangle \propto \phi_{nuc} \langle P \rangle$, and therefore a constant ratio of DNA and protein, $\langle D \rangle / \langle P \rangle \propto \phi_{nuc} / \lambda \propto const$. Instead, we observe a twofold change of DNA/protein (Fig. S14A), suggesting other factors limiting DNA synthesis.

However, because we cannot rule out a disproportionate compression of the abundance of some nucleotide synthesis enzymes under LacZ OE, we agree that our statement that "DNA synthesis is not nucleotide limited under LacZ OE" was somewhat speculative and we have therefore removed this statement in our revised manuscript.

We hope that this point has been clarified by the changes we made in our revised manuscript.

<Minor comments>

- It would be more informative to have per-cell values in Fig. S8

Per cell values are shown elsewhere: Fig. 1 and Fig. S6. The purpose of Fig. S8 (which becomes Fig. S9 in the revised manuscript) is to present the raw data which was normalized to OD.

- On pg. 6, 2nd paragraph, define 'D' earlier.

In our revised manuscript, the details of the quantitative discussion have been moved to the Supplementary Information. This quantity is now defined at the first mention of average cellular DNA and denoted by $\langle D \rangle$.

- In Fig. S3B, what are different symbols?

The different colored symbols represent growth on different carbon sources. We expanded the caption of the figure to clarify.

Thank you again for submitting your work to Molecular Systems Biology. We have now heard back from the referees who agreed to evaluate your manuscript. As you will see below, the referees are satisfied with the modifications made. However, reviewer #3 lists a few minor concerns, which we would ask you to address in a revision of the manuscript.

Moreover, we would ask you to address some editorial issues listed below.

Reviewer #2:

Due to other commitments I did not have the time to go through the manuscript in great detail. However, I read the manuscript and the authors response and thought the authors addressed the key concerns appropriately. I would have liked to have seen the inflation experiment done with a second protein, but can understand the authors' reluctance. In summary I think the paper is suitable for publication in MSB. I really like this kind of work and believe readers will too.

Reviewer #3:

The authors' responses have addressed most of my comments. However, I still have a couple of points that I believe need to be clarified before publication.

1. About the mean-field model. Now the authors introduced a population-averaged version of the threshold initiator model so that the model is consistent with experimentally-derived quantities. In the model, the authors defined $\phi_x = \langle P_x \rangle / \langle p \rangle$. Do the authors implicitly assume $\langle P_x / P \rangle = \langle P_x \rangle / \langle p \rangle$? Since at a single-cell level, $\phi_x = P_x / P$, so its population-average quantity should be $\langle P_x / P \rangle$, which is not necessarily the same as $\langle P_x \rangle / \langle p \rangle$.
2. In Fig. S12, isn't it more intuitive to show the $\phi_x \langle p \rangle$? I understand that the authors looked for the pattern shown in Fig. 2B, but ultimately what the threshold initiator model requires is $\phi_x \langle p \rangle$ being constant across different conditions, and this would be visually easy to assess.

Report for Author	The authors' responses have addressed most of my comments. However, I still have a couple of points that I believe need to be clarified before publication. 1. About the mean-field model. Now the authors introduced an population-averaged version of the threshold initiator model so that the model is consistent with experimentally-derived quantities. In the model, the authors defined $\phi_x = \langle P_x \rangle / \langle P \rangle$. Do the authors implicitly assume $\langle P_x / P \rangle = \langle P_x \rangle / \langle P \rangle$? Since at a single-cell level, $\phi_x = P_x / P$, so its population-average quantity should be $\langle P_x / P \rangle$, which is not necessarily the same as $\langle P_x \rangle / \langle P \rangle$. 2. In Fig. S12, isn't it more intuitive to show the $\phi_x \langle P \rangle$? I understand that the authors looked for the pattern shown in Fig. 2B, but ultimately what the threshold initiator model requires is $\phi_x \langle P \rangle$ being constant across different conditions, and this would be visually easy to assess.
--

We would like to thank the reviewer for his/her thorough analysis of our model. Our responses are given below.

1. The experimentally accessible quantity is the abundance of proteins as a fraction of the total proteome in the culture, denoted as φ_X . If N is the total cell number in the culture, then the total amount of the regulator X in the culture is given by $N \langle P_X \rangle$. On the other hand, the total amount of protein in the culture is given by $N \langle P \rangle$. The measurable relative abundance φ_X is given by $\varphi_X \equiv \frac{N \langle P_X \rangle}{N \langle P \rangle} = \frac{\langle P_X \rangle}{\langle P \rangle}$. The average cellular fractional abundance of X given by $\langle \frac{P_X}{P} \rangle$ does not play a role in our model, hence we make no assumptions regarding the equivalence of $\frac{\langle P_X \rangle}{\langle P \rangle}$ and $\langle \frac{P_X}{P} \rangle$.
2. We agree with the reviewer that this would be a good alternative way to analyze the proteomics data. Nevertheless, we prefer the presentation chosen currently to emphasize the qualitative similarities in the trends, rather than the constancy of their product. The latter is not perfect due to uncertainties in both measurements.